image processing

depth image, super-resolution, sparse code, joint trilateral filter

**Authors for correspondence:**
Dongsheng Zhou
e-mail: donyson@hotmail.com
Xin Yang
e-mail: xinyang@dlut.edu.cn

# Depth image super-resolution reconstruction based on a modified joint trilateral filter

Dongsheng Zhou[1], Ruyi Wang[1], Xin Yang[2], Qiang Zhang[1,2] and Xiaopeng Wei[2]

[1]Key Laboratory of Advanced Design and Intelligent Computing (Dalian University), Ministry of Education, Dalian 116622, People's Republic of China
[2]College of Computer Science and Technology, Dalian University of Technology, Dalian 116024, People's Republic of China

DZ, 0000-0003-3414-9623

Depth image super-resolution (SR) is a technique that uses signal processing technology to enhance the resolution of a low-resolution (LR) depth image. Generally, external database or high-resolution (HR) images are needed to acquire prior information for SR reconstruction. To overcome the limitations, a depth image SR method without reference to any external images is proposed. In this paper, a high-quality edge map is first constructed using a sparse coding method, which uses a dictionary learned from the original images at different scales. Then, the high-quality edge map is used to guide the interpolation for depth images by a modified joint trilateral filter. During the interpolation, some information of gradient and structural similarity (SSIM) are added to preserve the detailed information and suppress the noise. The proposed method can not only preserve the sharpness of image edge, but also avoid the dependence on database. Experimental results show that the proposed method is superior to some state-of-the-art depth image SR methods.

## 1. Introduction

The depth image is mainly used to record distance information from the camera to the objects in the scene. Such information is essential in some research fields, such as robot navigation [1], augmented reality [2], human pose estimation [3,4], hand pose estimation [5,6] and so on. Nowadays, depth image can be acquired easily using low-cost RGB-D sensors, such as Kinect cameras, PMD (photonic mixer device) cameras and so on [7]. Unfortunately, limited by the performance of those devices, the

resolution of acquired depth images is too low to meet the needs of many applications. To solve the above problems, the method for depth image super-resolution (SR) came into being.

Depth image SR is an important branch of image processing technology. In general, one or more low-resolution (LR) depth images will be chosen as the input and then mapped into a high-resolution (HR) image. Some prior information is essential when depth image is reconstructed. According to the prior information, depth image SR can be divided into four subclasses: (1) SR-based interpolation, (2) SR from LR depth image frames of the same scene, (3) example-based SR, (4) colour-guided SR. Different methods have different characteristics, including advantages and disadvantages.

In this paper, a modified joint trilateral filter is presented for depth image SR. Given an LR depth image, HR edge map is reconstructed first by the sparse coding method. Then, HR depth image is interpolated by joint trilateral filter with the guidance of HR edge. The proposed method has two main contributions: (i) The sparsity of edge map is used to reconstruct high-quality edges with self-similar patches without any external database. (ii) During the process of joint trilateral filtering, gradient information and structural similarity (SSIM) index are used to control depth interpolation.

The rest of this paper is organized as follows: In §2, the related works are briefly introduced. In §3, more details of the proposed method are described systematically. In §4, experiments and analyses are illustrated, especially the results of comparative experiments with some state-of-the-art methods. Finally, in §5, the conclusion of this paper is summarized, and problems and future work are presented.

## 2. Related works

In recent years, two major trends emerge in depth image SR. One is example-based depth image SR method. This method mainly reconstructs an HR depth image based on example databases that could be used to acquire learned prior information. For example, Aodha *et al.* [8] used the Markov random field (MRF) model-based patches for depth image SR. Li *et al.* [9] proposed a modified MRF model, which matched the input LR patches from similar patches on a set of HR training images. Besides, the approach based on sparse representation has also been used widely in depth image SR. Yang *et al.* [10] jointly trained the HR and LR dictionaries to enhance the coupling between HR and LR image blocks, which can be represented by an alternate atomic linear combination of the dictionaries. On the basis of sparse representation, Zhao *et al.* [11] proposed a multiresidue dictionary to learn and refine the depth image SR. Timofte *et al.* [12] clustered dictionary atoms into sub-dictionaries by using the K-NN algorithm and then represented the HR image blocks with the best sub-dictionary atoms. Owing to the effectiveness and speediness of neural networks in colour image processing, neural networks are also widely used in depth images. For example, Song *et al.* [13] used deep convolutional neural network to learn the end-to-end mapping from LR depth image to HR depth image, and then further process the learned HR depth images. Riegler *et al.* [14] proposed a depth image SR reconstruction method based on deep primal-dual networks, which combines a deep fully convolutional network with a non-local variation.

The other way is the colour-guided depth image SR method. RGB-D sensor can capture simultaneously depth image and the corresponding colour image, and the captured colour image usually has higher resolution than the depth image. Therefore, the colour image can be used to assist depth image SR. For example, Yang *et al.* [15] used one or two HR colour images as the reference, then refined the LR depth image iteratively. Ferstl *et al.* [16] used an anisotropic total variation diffusion tensor computed from the HR colour image to guide depth image SR. Lo *et al.* [17] proposed a framework of joint trilateral filter, the context information of which acquired from HR colour image was used to guide depth interpolation. Zhang *et al.* [18] presented a modified joint trilateral filter, and the depth image could be interpolated with the assistance of edge map and intensity information extracted from the HR colour image.

These two methods can improve the resolution of depth images, but there still exist some limitations. In general, the example-based SR method has a strong dependence on example database. And the colour-guided method requires HR colour images that are perfectly aligned with the depth images. To overcome these limitations, we propose a depth image SR method that needs neither the external example database nor the assistance of the registered HR colour image.

## 3. Methods

In this section, firstly, the whole framework of the proposed method is introduced. Then, the construction of high-quality edge is discussed. Finally, the modified joint trilateral filter is described that could be used to interpolate the depth image under the guidance of high-quality edge information.

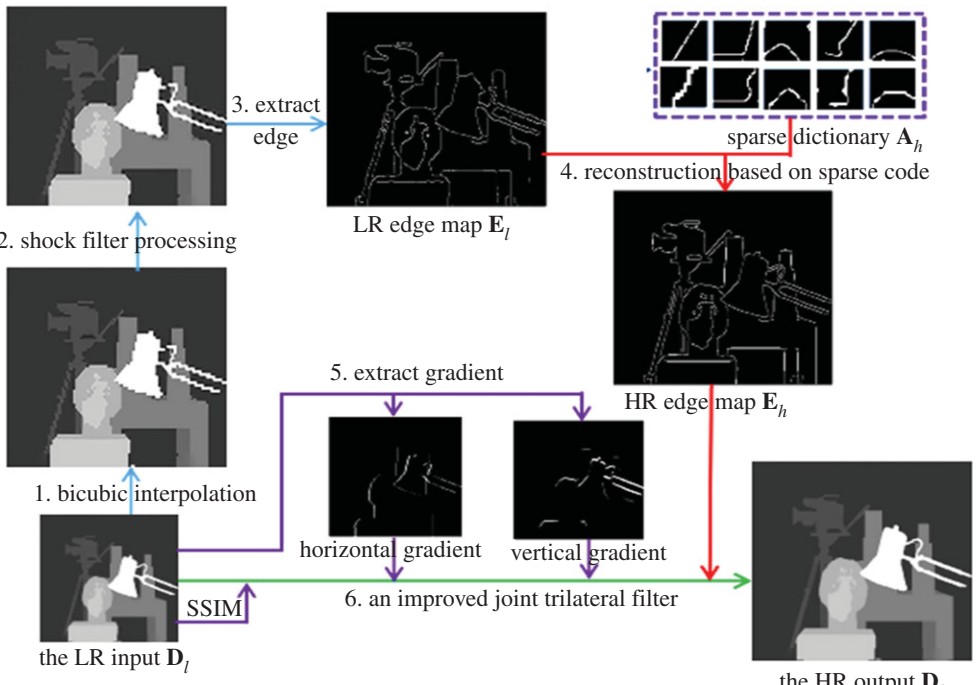

**Figure 1.** Whole framework of the proposed method.

The general steps of the proposed method are shown in figure 1. To keep sharp edge and overcome limitations of external database, a novel depth image SR method is presented, which employs a modified joint trilateral filter with edge guidance for LR-to-HR reconstruction.

As with the methods in [19], the input LR image $\mathbf{D}_l$ was firstly magnified to the same size as the expected HR image $\mathbf{D}_h$ by the bicubic interpolation algorithm. Then, a shock filter [20] was used to reduce jagged effects caused by interpolation algorithm and obtain depth image $\mathbf{D}'_l$.

Edge information is important for distinguishing different objects in the scene. So we first extracted edge map $\mathbf{E}_l$ from the preprocessed LR image $\mathbf{D}'_l$, then constructed high-quality edge map $\mathbf{E}_h$ from $\mathbf{E}_l$. Edge map has only some primary structure information made up of lines and angles which can lead to strong sparsity. So, the sparse coding method has the potential to recover high-quality edge maps. The sparse coding method, however, needs to train an over-complete dictionary from a set of images. Under the circumstance without external database, we constructed an edge map pyramid to find similar blocks for training, as shown in figure 2.

As far as the edge map is concerned, the larger its size is, the more self-similar blocks about edge and angle can be found. At the same time, self-similar blocks can be found more easily from the interpolated image of the test image than from the external image. These self-similar blocks can not only improve the efficiency of edge recovery, but also well retain the details of edge. We constructed the edge map pyramid based on the interpolated images of the test images at different scales. From figure 2, it can be seen that edge map pyramid can provide many self-similar blocks $\mathbf{P}'_1$ of block $\mathbf{P}_1$. Based on the extracted image blocks, an over-complete dictionary can be trained, and then edge map can be recovered by using the atoms of the over-complete dictionary.

Once the high-quality edge map is structured, depth image can be interpolated using a modified joint trilateral filter. Our modified trilateral filter can not only preserve the edge sharpness, but also further suppress the noise.

From the above overview, we can divide this method into two parts: (i) the construction of high-quality edge and (ii) edge-guided joint trilateral filter. More details will be introduced as follows.

## 3.1. The construction of high-quality edge

### 3.1.1. Dictionary training

In the beginning, the LR depth image $\mathbf{D}_l$ is the only original information. To obtain a dictionary training database, we constructed a pyramid of edge map. The process is as follows:

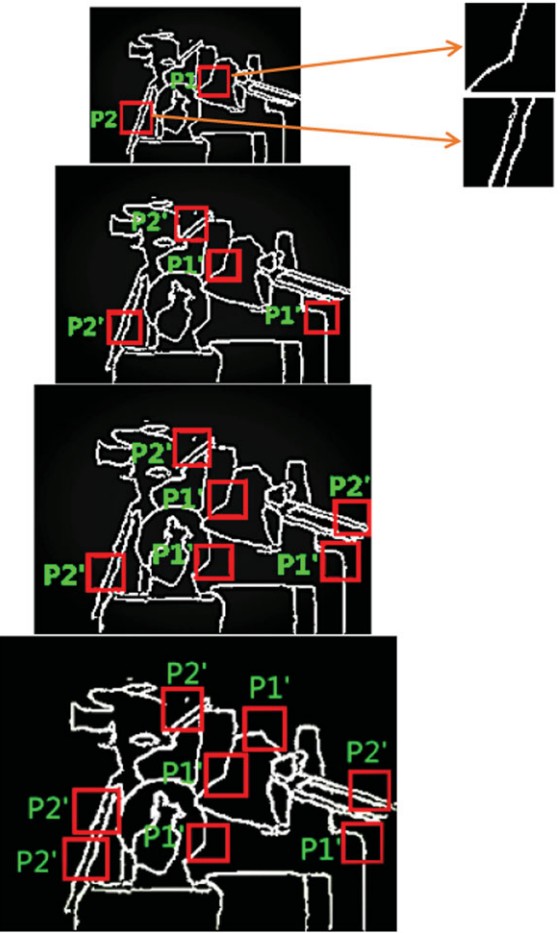

**Figure 2.** Edge map pyramid for searching similar blocks.

The LR depth image $\mathbf{D}_l$ is firstly interpolated by using the factors of $i$ ($i = 2,3,4$), and the interpolated images $\mathbf{D}_l^i$ can be generated. Then, edge maps $\mathbf{E}_l^0$ and $\mathbf{E}_l^i$ are extracted from depth images $\mathbf{D}_l$ and $\mathbf{D}_l^i$. Finally, a four-layer image pyramid can be constructed, which contains edge map $\mathbf{E}_l^0$ and $\mathbf{E}_l^i$, as shown in figure 2.

Then, image blocks of size $\sqrt{n} \times \sqrt{n}$ can be extracted from image pyramid and database $\{\mathbf{P}_k\}_j$ ($k$ is the index of image blocks, $j$ denotes the level of image pyramid) can be obtained. It can be seen that many blocks $\mathbf{P}_1'$ similar to $\mathbf{P}_1$ can be found, and some rough to fine information can be extracted from these similar blocks. A robust over-complete dictionary $\mathbf{A}_h \in \mathbb{R}^{nl \times nR}$ can be trained from database $\{\mathbf{P}_k\}_j$. For each image block $\mathbf{P}_k$, an alternative linear combination of its dictionary atoms can be found by using the K-SVD [21] algorithm:

$$\mathbf{A}_h, \{q^k\} = \arg\min_{\mathbf{A}_k} \sum_k \left\| \mathbf{P}_k - \mathbf{A}_h q^k \right\|_2^2 \quad \text{s.t.} \quad \left\| q^k \right\|_0 \le L \; \forall k \tag{3.1}$$

where $L$ is the sparse constraint, and $\{q^k\}_k$ is the sparse coding coefficient corresponding to the blocks $\{P_k\}_j$.

### 3.1.2. Edge map recovery

For the input LR depth image $\mathbf{D}_l$, it is firstly interpolated to the same size as the desired HR image $\mathbf{D}_h$. Then, a shock filter [20] is applied to eliminate jagged effects. Afterwards, edge map $\mathbf{E}_l$ is extracted from the processed image. HR edge map $\mathbf{E}_h$ can be recovered from $\mathbf{E}_l$. The detailed steps are as follows:

(1) Image blocks $\mathbf{b}_l^k$ ($k$ is the index of image blocks) of size $\sqrt{n} \times \sqrt{n}$ are extracted from edge map $\mathbf{E}_l$ at the location $k \in \Omega$;
(2) The corresponding HR blocks $\mathbf{b}_h^k$ can be represented by the sparse linear combinations of the atoms in the dictionary $\mathbf{A}_h$ using OMP [22] algorithm;

(3) The extracted blocks from the high-quality edge $\mathbf{E}_h$ should be as close as possible to $\mathbf{b}_h^k$. And the corresponding minimized cost function with respect to $\mathbf{E}_h$ is as follows:

$$\mathbf{E}_h = \arg\min_{\mathbf{E}_h} \sum_k \left\| \mathbf{R}_k \mathbf{E}_h - \mathbf{b}_h^k \right\|_2^2, \tag{3.2}$$

where $\mathbf{R}_k$ is the operator, which is used to extract image blocks with the same size $\sqrt{n} \times \sqrt{n}$ at the location $k \in \Omega$. HR edge map $\mathbf{E}_h$ can be acquired by using the least-squares approach.

## 3.2. Edge-guided joint trilateral filter

Once the high-quality edge map $\mathbf{E}_h$ is obtained, edge information will be used to guide the depth interpolation by using a modified joint trilateral filter. Each pixel $p$ in the expected SR depth image $\mathbf{D}_h$ can be derived as follows:

$$\mathbf{D}_h(p) = \frac{1}{k_p} \sum_{q \in \Omega} \mathbf{D}_l(q\!\downarrow) \cdot f_s(\| p\!\downarrow - q\!\downarrow \|) \cdot f_g(G_p - G_q) \cdot W_s \cdot f_r \, (\mathbf{E}_h, p, q), \tag{3.3}$$

where $k_p$ is a normalizing factor, $\Omega$ is a neighbourhood window centred at pixel $p$, pixel $q$ is the adjacent pixel of $p$ in the neighbourhood window, $f_s(\cdot)$ is the Gaussian function about spatial filter with standard deviation $\sigma_s$ and mean value 0, $f_g(\cdot)$ is the gradient Gaussian function of standard deviation $\sigma_g$ and mean value 0, which weighs the variation between pixel $p$ and pixel $q$, $W_s$ is the SSIM index, and $f_r(\cdot)$ is a function, which discriminates whether two pixels are at the same side of edge [23].

Based on the joint bilateral filter of Xie *et al.* [23], two constraint functions $f_g(\cdot)$ and $W_s$ about the spatial filter are added to preserve the detailed information. $f_g(\cdot)$ is used to compute the weight of pixel by gradient information. It is assumed that the coordinate of pixel $p$ is $(i, j)$ in image $\mathbf{D}_l$, and we computed firstly the abs of its first-order gradient $(G_v^1(i,j), G_h^1(i,j))$ on both vertical and horizontal directions,

$$G_v^1(i,j) = \left| \frac{\mathbf{D}_l\,(i+1,j) - \mathbf{D}_l\,(i-1,j)}{2} \right| \tag{3.4}$$

and

$$G_h^1(i,j) = \left| \frac{\mathbf{D}_l\,(i,j+1) - \mathbf{D}_l\,(i,j-1)}{2} \right|. \tag{3.5}$$

Two pixels may have the same gradient distribution near edge even if they are located on different depth planes. So, the second-order gradient is calculated to solve this problem,

$$G_v^2\,(i,j) = \frac{G_v^1\,(i+1,j) - G_h^1\,(i-1,j)}{2} \tag{3.6}$$

and

$$G_h^2\,(i,j) = \frac{G_h^1\,(i,j+1) - G_h^1\,(i,j-1)}{2}. \tag{3.7}$$

Then, $G_v^2\,(i,j)$ and $G_h^2\,(i,j)$ will be used as the input of two-dimensional Gauss distribution to compute the weight between adjacent pixels.

The SSIM index [24] $W_s$ is used to relieve the impact of noise. $W_s$ is composed of three parts, including the mean function $m(p,q)$, the standard deviation function $\sigma(p,q)$ and the structure comparison function $s(p,q)$ that is conducted on the normalizing signals $p - \mu_p/\sigma_p$ and $q - \mu_p/\sigma_q$.

$$m(p,q) = \frac{2\mu_p\mu_q + C_1}{\mu_p^2 + \mu_q^2 + C_1}, \tag{3.8}$$

$$\sigma(p,q) = \frac{2\sigma_p\sigma_q + C_2}{\sigma_p^2 + \sigma_q^2 + C_2} \tag{3.9}$$

and

$$s(p,q) = \frac{2\sigma_{pq} + C_3}{\sigma_p\sigma_q + C_3}, \tag{3.10}$$

where $C_1$, $C_2$ and $C_3$ are non-zero constants, which are used to avoid zero denominator. $\mu_p$ and $\sigma_p$ are the mean and the standard deviation of the pixels, respectively, in the neighbourhood window centred at pixel $p$. Furthermore, $\mu_q$ and $\sigma_q$ denote the mean and the standard deviation of the pixels, respectively, in the

**Table 1.** The parameter settings.

| parameters | $n$ | $s$ | $\sigma_s$ | $\sigma_g$ | $C_1$ | $C_2$ | $C_3$ | $\alpha$ | $\beta$ | $\gamma$ |
|---|---|---|---|---|---|---|---|---|---|---|
| values | 9 | 7 | 0.5 | 0.5 | 6.5 | 58.5 | 29.3 | 1 | 1 | 1 |

neighbourhood window centred at pixel $q$. $\sigma_{pq}$ is the covariance of two neighbourhoods centred at pixel $p$ and $q$. So, the SSIM index $W_s$ can be expressed as follows:

$$W_s = \mathrm{SSIM}(p,q) = m(p,q)^{\alpha} \sigma(p,q)^{\beta} \mathrm{s}(p,q)^{\gamma} \tag{3.11}$$

where $\alpha$, $\beta$ and $\gamma$ are weight factors.

# 4. Experiment and discussion

In this section, experimental environment and parameter settings are introduced. Then, the comparisons between the proposed method and four state-of-the-art methods in terms of quality and quantity are illustrated and analysed.

## 4.1. Experimental environment and parameter setting

In the experiments, we conducted the simulations on Matlab 2016a. The configuration of computer is Intel(R) Xeon(R) E5–2620 v3@ 2.40 Hz CPU and 64.0 GB RAM. The test images come from the Middlebury Stereo database [25,26]. For parameters $C_1$, $C_2$, $C_3$, $\alpha$, $\beta$ and $\gamma$, their values were selected by the default values of the structural similarity index (SSIM). Based on the papers of Yang *et al.* [10] and Xie *et al.* [23], an initial value was given to parameters of $n$, $s$, $\sigma_s$ and $\sigma_g$. Then, we computed the root mean square error (RMSE) with one parameter changing at a time and all the others constant until the average RMSE of all test images reached their minimum. Finally, these parameters were determined via trial-and-error. The values of parameters are listed in table 1.

## 4.2. Performance evaluation

In this subsection, LR test images were reconstructed by four state-of-the-art methods and the proposed method. The RMSE, the peak signal noise ratio (PSNR), the structural similarity (SSIM) and the percentage error (PE) were chosen as the assessment measures to evaluate the reconstructed results. As suggested in [22], PE is the percentage of the absolute difference in disparity that exceeds 1.

### 4.2.1. Methods of comparison

Four compared methods were provided in our experiments and carried out under the same condition. These compared methods include adjusted anchored neighbourhood regression for fast super-resolution (AANR) of Timofte *et al.* [12], accurate image super-resolution using very deep convolutional networks (CNN) of Kim *et al.* [27], the modified sparse coding method of Zeyde *et al.* [19] and the edge-guided method of Xie *et al.* [23].

### 4.2.2. Analysis of experimental results

As for the input LR test images, we obtained them by down-sampling the ground truth HR counterparts. Then, LR test images were reconstructed by the proposed method and four compared methods. To demonstrate the validity of the proposed method, we evaluated the reconstructed results of 4× scaling factor by the above four assessment measures. The experimental results are shown in tables 2–5.

The top two best SR methods are marked in tables 2–5. The values in bold indicate the best results. The values in italics indicate the second best results. It can be seen from tables 2 and 4 that, both the RMSE values and the PSNR values of the proposed method ranked the first among the compared methods. In tables 3 and 5, we can see that the SSIM and PE values of the proposed method ranked the top two in all test results.

**Table 2.** RMSE values on the Middlebury Stereo database with scaling factor of 4.

| RMSE ×4 | bowling | aloe | cones | Indian | Venus | warrior | tsukuba | hand | dove |
|---|---|---|---|---|---|---|---|---|---|
| Timofte | 1.855 | 2.478 | 1.456 | 0.855 | 0.674 | 3.707 | 2.972 | 1.925 | 1.043 |
| Kim | 2.238 | 3.245 | 1.778 | 0.987 | 0.845 | 4.424 | 3.505 | 2.174 | 1.214 |
| Zeyde | 1.803 | *2.329* | 1.338 | 0.798 | 0.635 | *3.620* | *2.844* | 1.832 | *0.989* |
| Xie | *1.766* | 2.583 | *1.240* | *0.771* | *0.617* | 4.081 | 3.009 | *1.926* | 1.010 |
| ours | **1.623** | **2.217** | **1.214** | **0.703** | **0.553** | **3.316** | **2.638** | **1.654** | **0.919** |

**Table 3.** SSIM values on the Middlebury Stereo database with scaling factor of 4.

| SSIM ×4 | bowling | aloe | cones | Indian | venus | warrior | tsukuba | hand | dove |
|---|---|---|---|---|---|---|---|---|---|
| Timofte | 0.924 | 0.880 | 0.891 | 0.987 | 0.953 | 0.906 | 0.801 | 0.985 | 0.990 |
| Kim | 0.922 | 0.865 | 0.880 | 0.987 | 0.951 | 0.904 | 0.843 | 0.983 | 0.988 |
| Zeyde | 0.925 | 0.885 | 0.893 | 0.988 | 0.950 | 0.905 | 0.839 | 0.984 | 0.989 |
| Xie | *0.946* | *0.908* | *0.916* | *0.992* | **0.971** | *0.931* | *0.855* | **0.989** | **0.993** |
| ours | **0.962** | **0.921** | **0.919** | **0.993** | *0.969* | **0.938** | **0.882** | *0.987* | *0.992* |

**Table 4.** PSNR values on the Middlebury Stereo database with scaling factor of 4.

| PSNR ×4 | bowling | aloe | cones | Indian | venus | warrior | tsukub | hand | dove |
|---|---|---|---|---|---|---|---|---|---|
| Timofte | 42.761 | 40.245 | 44.864 | 49.485 | 51.553 | 36.748 | 38.669 | 42.451 | 47.762 |
| Kim | 40.667 | 37.764 | 43.131 | 48.237 | 49.587 | 35.214 | 37.237 | 41.396 | 46.444 |
| Zeyde | *43.008* | 40.784 | 45.599 | 50.085 | 52.071 | *36.954* | *39.049* | *42.882* | *48.219* |
| Xie | 42.124 | 39.332 | *46.260* | *50.384* | *52.317* | 35.915 | 38.560 | 42.447 | 48.044 |
| ours | **43.312** | **41.217** | **46.339** | **51.206** | **53.531** | **37.745** | **39.705** | **43.553** | **48.932** |

**Table 5.** PE values on the Middlebury Stereo database with scaling factor of 4.

| PE ×4 | bowling | aloe | cones | Indian | venus | warrior | tsukuba | hand | dove |
|---|---|---|---|---|---|---|---|---|---|
| Timofte | 5.274 | 14.741 | 7.385 | 2.047 | 1.967 | 7.640 | 12.816 | 3.293 | 2.248 |
| Kim | 4.232 | 13.454 | 6.993 | 1.795 | 1.696 | 8.077 | 11.340 | 2.624 | 1.690 |
| Zeyde | 6.040 | 15.751 | 7.968 | 2.208 | 2.447 | 8.228 | 14.240 | 4.036 | 2.544 |
| Xie | *2.405* | *8.299* | **2.829** | *0.951* | **0.505** | *2.575* | *4.239* | *0.918* | **0.608** |
| ours | **2.365** | **8.242** | *3.154* | **0.943** | *0.641* | *2.593* | **4.217** | **0.912** | *0.724* |

To evaluate the performance in qualitative sense, in figures 3 and 4, we provide the ground-truth HR image of test image 'bowling' and 'dove' and their reconstructed images (4× scaling factor), respectively. From these images, it can be observed that our reconstructed depth images can not only avoid blurred edges, but also help reduce zigzags near edges.

# 5. Conclusion and future work

In this paper, a novel depth image SR method is proposed that does not need the assistance of any external images. To avoid blurred and jagged results on the edge of the final image, we first

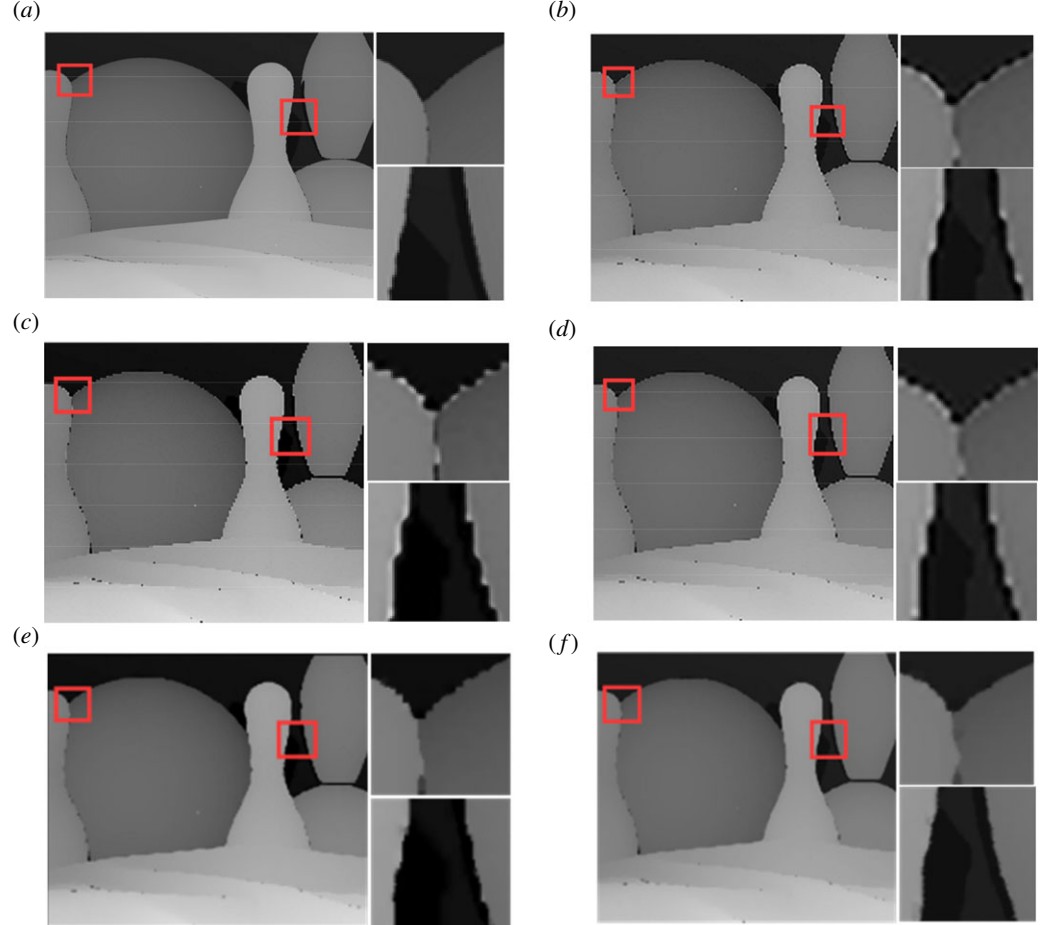

**Figure 3.** Comparison of 'bowling' with two regions of interest. (*a*) Ground truth, (*b*) Timofte [12], (*c*) Kim [27], (*d*) Zeyde [19], (*e*) Xie [23], (*f*) the proposed.

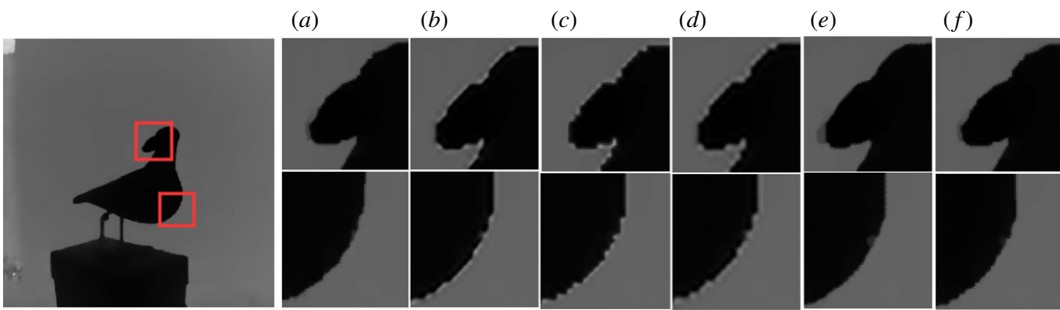

**Figure 4.** Comparison of 'dove' with two regions of interest. (*a*) Ground truth, (*b*) Timofte [12], (*c*) Kim [27], (*d*) Zeyde [19], (*e*) Xie [23], (*f*) the proposed.

reconstructed high-quality edge map by the sparse coding method. What differs from other sparse coding methods is that our sparse dictionary is trained from the interpolated images of the LR test image at different scales. Then, under the guidance of high-quality edge, depth image was interpolated by a modified trilateral filter. We applied local gradient information and SSIM index to preserve detailed information and suppress noise when interpolation was performed. Quantitative and qualitative experimental analyses demonstrate that the proposed method can obtain better results than some state-of-the-art methods.

However, there still exist shortages in the proposed method. Running time of the proposed method is higher than other methods because this method needs to construct dataset and train sparse dictionary

**Table 6.** Running time on the Middlebury Stereo database with scaling factor 4.

| Time (s) | bowling | aloe | cones | Indian | venus | warrior | tsukuba | hand | dove |
|---|---|---|---|---|---|---|---|---|---|
| Timofte | 3.4 | 2.0 | 1.6 | 6.4 | 1.6 | 4.5 | 0.9 | 6.4 | 6.2 |
| Kim | 6.7 | 4.8 | 3.5 | 12.9 | 3.6 | 10.0 | 2.3 | 13.7 | 12.9 |
| Zeyde | 5.3 | 3.1 | 2.7 | 9.9 | 2.4 | 6.7 | 1.3 | 9.9 | 10.6 |
| Xie | 594.9 | 864.6 | 608.9 | 913.7 | 141.3 | 759.9 | 373.8 | 469.5 | 417.7 |
| ours | 95.3 | 116.7 | 121.4 | 195.4 | 73.9 | 135.2 | 71.2 | 93.4 | 91.7 |

during depth image SR (table 6). And the process of choosing parameters is complicated. In the future, we will further improve the works as follows: (i) Edge recovery: we will recover HR edge map with an effective method. (ii) Parameter setting: a graphical user interface (GUI) will be designed to choose parameters as shown in table 1.

Data accessibility. The codes and data are deposited at the Dryad Digital Repository: http://dx.doi.org/10.5061/dryad.5ph7sm6 [28].

Authors' contributions. R.W. wrote this manuscript; D.Z., X.Y., Q.Z. and X.W. equally contributed to the writing, direction, content and also revised the manuscript.

Competing interests. The authors declare no conflict of interest.

Funding. This work is supported by the National Natural Science Foundation of China (nos. 61751203, 61603066 and 91748104), Program for the Liaoning Distinguished Professor, Program for Dalian High-level Talent's Innovation (2015R088), Innovation Fund Plan for Dalian Science and Technology (2018J12GX036) and Program for Changjiang Scholars and Innovative Research Team in University (no. IRT_15R07).

Acknowledgements. The authors acknowledge the reviewer's valuable suggestions.

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
