## [Reviewer comments · Royal Society Open Science]

Review History

RSOS-181074.R0 (Original submission)

Review form: Reviewer 1

Is the manuscript scientifically sound in its present form?

Yes

Are the interpretations and conclusions justified by the results?

Yes

Is the language acceptable?

Yes

Is it clear how to access all supporting data?

Yes

Do you have any ethical concerns with this paper?

No

Have you any concerns about statistical analyses in this paper?

No

Recommendation?

Accept with minor revision (please list in comments)

Comments to the Author(s)

At present, with the popularity of deep camera systems, such as Microsoft's kinect, Apple's iphoneX and other consumer devices have already adopted a deep camera system, but the common problem is low precision. Based on the above background, this paper has carried out research on the super-resolution enhancement method of depth image, which has good frontier and practicality. This paper proposes a depth image enhancement method based on an improved joint trilateral filter. More interesting is that this manuscript proposes a method for super-resolution image reconstruction without relying on external databases or high-resolution images to obtain prior information. The overall logical framework of the paper is clear and the overall writing is better. The method description, demonstration and experiments are complete, and the results are credible. But, some of the following questions should be considered:

1. I recommended having a native English speaker to go through this paper.
2. There are many spelling errors in the paper. For example, the characters DI on the 5th line and the 8th line of page 4 are garbled. Please modify
3. There is an error in the chapter number of the paper. For example, there exists inconsistency in the subsection numbers in Sections 4 and 5.
4. Table 1 lists the 10 parameter values set before the experiment. How these parameter values are determined should be explained in detail.
5. There are some bugs in the running of the attached program code. It is recommended to attach a detailed program running instructions.

Generally, the manuscript is good with nice motivations, method and extensive experiments. I recommend accepting this paper after minor revision.

Review form: Reviewer 2 (Kaichun Zhao)

Is the manuscript scientifically sound in its present form?

Yes

Are the interpretations and conclusions justified by the results?

Yes

Is the language acceptable?

Yes

Is it clear how to access all supporting data?

Yes

Do you have any ethical concerns with this paper?

No

Have you any concerns about statistical analyses in this paper?

No

Recommendation?

Accept with minor revision (please list in comments)

Comments to the Author(s)

The work at hand proposes an interesting method for deep image SR which needs neither the external example database nor the assistance of the registered HR color image. The core of the proposed method was named Edge Guided Joint Trilateral Filter. The whole paper is clear, well written, and easy to follow the motivation. The theory is correctly derived and the experimental analysis is sufficient. Overall I am not convinced of this manuscript in its current form.

1. The number of the references needs to be reorder. In Fig.3, the "Timofe [11]" is should be [12].....
2. Overall the quality of the Figures is not satisfying. For example, the Fig.2 should be included as vector graphics or in an adequate resolution, such that details are not covered by compression artifacts. Or, maybe, the low resolution pictures which look not clear are waiting for SR?
3. Is there any discussion about table 6, why the running time of the proposed method is higher than most other methods?

Decision letter (RSOS-181074.R0)

24-Oct-2018

Dear Professor Zhou,

The editors assigned to your paper ("Deep Image Super-Resolution Reconstruction Based on an Improved Joint Trilateral Filter") have now received comments from reviewers. We would like you to revise your paper in accordance with the referee and Associate Editor suggestions which can be found below (not including confidential reports to the Editor). Please note this decision does not guarantee eventual acceptance.

Please submit a copy of your revised paper before 16-Nov-2018. Please note that the revision deadline will expire at 00.00am on this date. If we do not hear from you within this time then it will be assumed that the paper has been withdrawn. In exceptional circumstances, extensions may be possible if agreed with the Editorial Office in advance. We do not allow multiple rounds of revision so we urge you to make every effort to fully address all of the comments at this stage. If deemed necessary by the Editors, your manuscript will be sent back to one or more of the original reviewers for assessment. If the original reviewers are not available, we may invite new reviewers.

- Data accessibility

If you wish to submit your supporting data or code to Dryad (<http://datadryad.org/>), or modify your current submission to dryad, please use the following link:
<http://datadryad.org/submit?journalID=RSOS&manu=RSOS-181074>

- Competing interests

- Authors' contributions

- Acknowledgements

- Funding statement

Please note that Royal Society Open Science charge article processing charges for all new submissions that are accepted for publication. Charges will also apply to papers transferred to Royal Society Open Science from other Royal Society Publishing journals, as well as papers submitted as part of our collaboration with the Royal Society of Chemistry (<http://rsos.royalsocietypublishing.org/chemistry>). If your manuscript is newly submitted and subsequently accepted for publication, you will be asked to pay the article processing charge, unless you request a waiver and this is approved by Royal Society Publishing. You can find out more about the charges at <http://rsos.royalsocietypublishing.org/page/charges>. Should you have any queries, please contact openscience@royalsociety.org.

on behalf of Prof. Marta Kwiatkowska (Subject Editor)
openscience@royalsociety.org

Comments to Author:

Reviewers' Comments to Author:
Reviewer: 1

Comments to the Author(s)

At present, with the popularity of deep camera systems, such as Microsoft's kinect, Apple's iPhone X and other consumer devices have already adopted a deep camera system, but the common problem is low precision. Based on the above background, this paper has carried out research on the super-resolution enhancement method of depth image, which has good frontier and practicality. This paper proposes a depth image enhancement method based on an improved joint trilateral filter. More interesting is that this manuscript proposes a method for super-resolution image reconstruction without relying on external databases or high-resolution images to obtain prior information. The overall logical framework of the paper is clear and the overall writing is better. The method description, demonstration and experiments are complete, and the results are credible. But, some of the following questions should be considered:

1. I recommended having a native English speaker to go through this paper.
2. There are many spelling errors in the paper. For example, the characters DI on the 5th line and the 8th line of page 4 are garbled. Please modify
3. There is an error in the chapter number of the paper. For example, there exists inconsistency in the subsection numbers in Sections 4 and 5.
4. Table 1 lists the 10 parameter values set before the experiment. How these parameter values are determined should be explained in detail.
5. There are some bugs in the running of the attached program code. It is recommended to attach a detailed program running instructions.

Generally, the manuscript is good with nice motivations, method and extensive experiments. I recommend accepting this paper after minor revision.

Reviewer: 2

Comments to the Author(s)

The work at hand proposes an interesting method for deep image SR which needs neither the external example database nor the assistance of the registered HR color image. The core of the proposed method was named Edge Guided Joint Trilateral Filter. The whole paper is clear, well written, and easy to follow the motivation. The theory is correctly derived and the experimental analysis is sufficient. Overall I am not convinced of this manuscript in its current form.

1. The number of the references needs to be reorder. In Fig.3, the "Timofe [11]" is should be [12].....
2. Overall the quality of the Figures is not satisfying. For example, the Fig.2 should be included as vector graphics or in an adequate resolution, such that details are not covered by compression artifacts. Or, maybe, the low resolution pictures which look not clear are waiting for SR?
3. Is there any discussion about table 6, why the running time of the proposed method is higher than most other methods?

Author's Response to Decision Letter for (RSOS-181074.R0)

See Appendices A and B.

RSOS-181074.R1 (Revision)

Review form: Reviewer 1

Is the manuscript scientifically sound in its present form?

Yes

Are the interpretations and conclusions justified by the results?

Yes

Is the language acceptable?

Yes

Is it clear how to access all supporting data?

Yes

Do you have any ethical concerns with this paper?

No

Have you any concerns about statistical analyses in this paper?

No

Recommendation?

Accept as is

Comments to the Author(s)

No. The authors have revised the manuscript based on the comments. The quality of pictures should be further improved.

Review form: Reviewer 2 (Kaichun Zhao)

Is the manuscript scientifically sound in its present form?

Yes

Are the interpretations and conclusions justified by the results?

Yes

Is the language acceptable?

Yes

Is it clear how to access all supporting data?

Yes

Do you have any ethical concerns with this paper?

No

Have you any concerns about statistical analyses in this paper?

Yes

Recommendation?

Accept as is

Comments to the Author(s)

Your research job is valuable and interesting.

Decision letter (RSOS-181074.R1)

17-Dec-2018

Dear Professor Zhou,

I am pleased to inform you that your manuscript entitled "Depth Image Super-Resolution Reconstruction Based on a Modified Joint Trilateral Filter" is now accepted for publication in Royal Society Open Science.

on behalf of Prof Marta Kwiatkowska (Subject Editor)
openscience@royalsociety.org

Associate Editor Comments to Author:

Thank you for supplying this revision. The referees are broadly content that your paper is ready for acceptance -- the Editors would, however, like you to bear in mind the comments from one of the reviewers that the figures in the manuscript need to be clarified. Please ensure this is addressed during typesetting and proofing. Thank you for your support of Royal Society Open Science.

Reviewer comments to Author:
Reviewer: 1

Comments to the Author(s)
No. The authors have revised the manuscript based on the comments. The quality of pictures should be further improved.

Reviewer: 2

Comments to the Author(s)
Your reseach job is valuable and intersting.

Appendix A

Explanation of this revision

Thank you very much for your valuable comments, which has been very useful for improving the quality of this paper. The details of the revision are as follows:

Question: 1

I recommended having a native English speaker to go through this paper.

Answer:

Thank you for your careful reading very much. The whole manuscript has been reedited by a native English speaker who fixed many grammar and expression errors.

Question:2

There are many spelling errors in the paper. For example, the characters D_i' on the 5th line and the 8th line of page 4 are garbled. Please modify

Answer:

The characters D_i' have been modified on the 4th line and the 6th line of page 3, and further, we also examined the entire manuscript and eliminated the similar problem.

Question:3

There is an error in the chapter number of the paper. For example, there exists inconsistency in the subsection numbers in Sections 4 and 5.

Answer:

Thank you for your reminder. We have reedited the subsection numbers in Sections 4 and 5.

Question:4

Table 1 lists the 10 parameter values set before the experiment. How these parameter values are determined should be explained in detail.

Answer:

The selection of parameters is introduced on the lines 26-30 of page 5 in detail.

Question:5

There are some bugs in the running of the attached program code. It is recommended to attach a detailed program running instructions.

Answer:

The new version of the codes has been uploaded to the Dryad website, which added the program description, and some bugs have been fixed. You can download the new codes from the URL: <https://datadryad.org/review?doi=doi:10.5061/dryad.5ph7sm6>

Appendix B

Explanation of this revision

Thank you very much for your valuable comments which has been very useful for improving the quality of this paper. The details of the revision are as follows:

Question: 1

The number of the references needs to be reorder. In Fig.3, the “Timofe [11]” is should be [12].....

Answer:

Thank you for your careful reading very much. The numbers of the references have been reordered in Fig.3 and Fig.4.

Question: 2

Overall the quality of the Figures is not satisfying. For example, the Fig.2 should be included as vector graphics or in an adequate resolution, such that details are not covered by compression artifacts. Or, maybe, the low resolution pictures which look not clear are waiting for SR?

Answer:

The original figures are very large, so what show in Fig.1 and Fig.2 are compressed screenshots. High resolution images have replaced the unclear images.

Question: 3

Is there any discussion about table 6, why the running time of the proposed method is higher than most other methods?

Answer:

Thanks very lot for your reminder. The proposed method needs construct dataset and train sparse dictionary before depth image SR, so the running time is higher. And the explanation has been added on the lines 13-14 of page 8 in the manuscript.